# An examination of difficulties accessing surgical care in Canada from 2005-2014: Results from the Canadian Community Health Survey

Jordana Liyat Sommer[1,2], Edward Noh[3], Eric Jacobsohn[1], Chris Christodoulou[1], Renée El-Gabalawy[1,2,4,5]*

1 Department of Anesthesiology, Perioperative and Pain Medicine, University of Manitoba, Winnipeg, Manitoba, Canada, 2 Department of Psychology, University of Manitoba, Winnipeg, Manitoba, Canada, 3 Rady Faculty of Medicine, University of Manitoba, Winnipeg, Manitoba, Canada, 4 Department of Clinical Health Psychology, University of Manitoba, Winnipeg, Manitoba, Canada, 5 Department of Psychiatry, University of Manitoba, Winnipeg, Manitoba, Canada

* Renee.El-Gabalawy@umanitoba.ca

**Data Availability Statement:** Data from this study cannot be shared publicly. Restrictions are in place to protect the privacy of survey respondents. Data

## Abstract

### Background

Difficulties accessing surgical care (e.g., related to wait times, cancellations, cost, receiving a diagnosis) are understudied in Canada. Using population-based data, we studied difficulty accessing non-emergency surgical care, including (1) the incidence and annual changes in incidence, (2) types of difficulties, and (3) associated factors (e.g., sociodemographics, surgery characteristics).

### Methods

Cross-sectional data from the Canadian Community Health Survey annual components were analyzed from 2005–2014. Weighted frequencies established the annual incidence of difficulty accessing surgical care, and total incidence of types of difficulties. Chi-square analyses, independent samples t-tests, and a multivariable logistic regression examined sociodemographic and surgery-related characteristics associated with difficulty accessing surgical care.

### Results

Among individuals who required past-year non-emergency surgery between 2005–2014 (weighted $n = 3,052,072$), 15.6% experienced difficulty accessing surgical care. The most common difficulty was "waited too long for surgery" (58.5%). There were significant differences in the incidence of difficulty according to year ($X^2 = 83.50$, $p < .001$) from 2005–2014. The incidence of difficulty accessing surgery varied according to sex ($X^2 = 4.02$, $p < .05$), surgery type ($X^2 = 96.09$, $p < .001$), party responsible for cancellation/postponement ($X^2$ range: 4.36–19.01, $p < .05$), and waiting time ($t = 10.59$, $p < .001$). In particular, males,

are protected through Statistics Canada, and researchers may request access through their Research Data Centre (https://www.statcan.gc.ca/eng/microdata/data-centres/access). Some elements of the survey are accessible online (see CCHS public use micro data files). The authors had no special access privileges to the data others would not have.

**Funding:** This work was supported by University of Manitoba Start-Up Funding (El-Gabalawy) and Max Rady College of Medicine and Rady Faculty of Health Sciences, University of Manitoba BSc(Med) program.

**Competing interests:** The authors have declared that no competing interests exist.

orthopedic surgery, and surgery cancelled by the surgeon or hospital had the highest rates of difficulty.

## Conclusion

Results provide insight into the difficulties experienced by patients accessing elective surgery, and the associated factors. These results may inform targeted healthcare interventions and resource reallocation to reduce these occurrences.

## Introduction

There are worldwide difficulties for patients in accessing elective surgery [1–9], and an estimated 5–44% of patients experience surgical cancellations [4–7,10]. However, these rates greatly vary from country to country. According to the Fraser Institute's National Waiting List Survey, Canadians often experience longer wait times for elective surgery than is considered "clinically reasonable" [2,3]; for example, orthopedic surgery patients are required to wait nearly 12 weeks longer than is "clinically reasonable" (23.8 weeks versus 12 weeks [3]).

Difficulties accessing surgical care are problematic as they are associated with increased morbidity, including impaired mobility, increased pain, reduced quality of life, deterioration in general health status, greater length of hospital stay, and increased mortality [11–13]. The implications of cancellations for health care organizations include increased hospital expenditures, reduced staff morale, and decreased operating room efficiency (e.g., greater unused operating room opportunities [11,14–16]). Several strategies have been employed to reduce the frequency of these difficulties, including increased health services funding, human resource policies in health care organizations, and modifications to pre-surgical screening and scheduling procedures [17]. However, despite these efforts, frequent short-notice cancellations and lengthy waiting times persist in Canada [1–3].

While some studies have estimated rates of difficulties accessing surgical care [3–6], they have largely focused on cancellations and wait times. For example, although the Fraser Institute releases annual data on waiting times for surgery (and other health services) across Canada [2,3], this research initiative does not investigate other types of difficulties associated with accessing surgical care, nor does it identify which subgroups among the population are most commonly affected by these difficulties. Similar to the Fraser Institute reports, extant research is lacking on other difficulties including issues with scheduling appointments, receiving a diagnosis, transportation, and cost, among others. Furthermore, although certain reasons for specific difficulties, such as cancellations, have been explored (e.g., hospital bed or operating room unavailability, changes in patient medical status [5–8,10]), there is limited research understanding sociodemographic characteristics associated with experiencing difficulties. It is essential to understand which subpopulations are most greatly affected by these difficulties so that targeted strategies can be developed for equal healthcare opportunities. Finally, though population-based estimates of surgical cancellations have been assessed in other countries [4,6], to our knowledge there is no research that has established a Canadian population-based estimate of cancellations or other difficulties, or that has examined changes in estimates over time.

Using Canadian population-based data, the aims of this study are to examine difficulty accessing non-emergency surgical care, including: 1) the annual and overall incidence, and changes in incidence over approximately a decade; 2) types of difficulties experienced; and 3)

person-specific (e.g., sex, age) and surgery-related (e.g., type of surgery, wait time) factors associated with difficulty accessing surgical care.

## Materials and methods

### Sample

Data were extracted from the annual components of the Canadian Community Health Survey (CCHS) for the years 2005–2014. The CCHS is an annual cross-sectional population-based survey conducted by Statistics Canada. It is administered annually to a nationally representative sample of approximately 65,000 Canadians, aged 12 years and older (prior to 2007, the data were collected every two years). Trained lay interviewers, using computer-assisted software, interview consenting participants. The exclusion criteria include Canadians living on reserves, Canadian Armed Forces members, and institutionalized individuals. The response rates range from 65.6%-78.9%, depending on the year (see S1 Appendix for annual response rates). The primary variables for the current study were drawn from the CCHS access to health care services (ACC) and waiting times (WTM) modules. These modules were optional, allowing provinces to choose to opt out from participating (see S1 Table). Only individuals who reported that they required non-emergency surgery in the past-year were included in primary analyses. In order to ensure representation of the population, the data were weighted annually using Statistics Canada weights, based on census data for that year. The full details regarding the survey methodology and sampling procedures have been published elsewhere [18].

This study followed the Strengthening the Reporting of Observational Studies in Epidemiology (STROBE) Statement guidelines. In order to access these data at the Manitoba Research Data Centre, this study received clearance from Statistics Canada, and was exempt from institutional ethical approval due to Statistics Canada's procedures for mitigating risk to respondents [19].

### Measures

**Non-emergency surgery and difficulty accessing surgical care.**    As part of the CCHS module on access to health care services (ACC), past-year non-emergency surgery was assessed by self-report to a single item included in the survey: "In the past 12 months, did you require any non-emergency surgery?"; this included both participants who received their surgery, and those who are still waiting for their surgery. If participants answered "yes", they were subsequently asked: "In the past 12 months, did you ever experience any difficulties getting the surgery you needed?" (i.e., any difficulty). If they had answered "yes" to this, they were then asked: "What types of difficulties did you experience?". Participants were provided with a list of response options, and were permitted to endorse multiple types of difficulties. We categorized the reasons for difficulties in the following groups: physician/systemic reasons (i.e., waited too long for surgery, difficulty getting an appointment, waited too long for a diagnostic test, still waiting, difficulty getting a diagnosis, waited too long for a hospital bed, appointment cancelled/deferred, service not available); patient reasons (i.e., deterioration of health, cost, difficulty acquiring transportation, unable to leave house due to health, personal/family responsibility, language barrier); and other reasons.

**Sociodemographics.**    We assessed age continuously, and categorized sex (male, female), household income ($29,999 or less, $30,000–59,999, $60,000+), racial origin (White, Other), urbanicity (urban, rural), and marital status (married/common-law, widowed/separated/ divorced, single), in accordance with prior research [20,21].

**Surgery-related characteristics.**    As part of the waiting times (WTM) module, respondents who required a past-year non-emergency surgery were subsequently asked: 1) the type

of surgery (i.e., cardiac, cancer, hip/knee, cataract/eye, hysterectomy, gall bladder, other [i.e., any other surgery that is not already specified]); 2) the party responsible for cancelling or postponing the surgery (if applicable; i.e., yourself, the surgeon, the hospital [no other alternatives, such as anesthesiologist, were provided]); and 3) whether their surgery required an overnight hospital stay (yes/no).

**Waiting times.** Also as part of the WTM module, respondents who reported requiring a past-year non-emergency surgery were asked how long they waited for surgery, whether or not they thought this was an acceptable waiting time, and what they would consider an acceptable waiting time in their particular case. We converted all waiting times to days (see S2 Appendix for specific waiting times questions).

## Analytic strategy

The protected health data were analyzed at the Research Data Centre in Winnipeg, Manitoba. Weighted frequencies established the incidence of past-year non-emergency surgery. Among those individuals who required a non-emergency surgery, weighted frequencies established the annual incidence of difficulty accessing surgical care. A chi-square analysis determined whether the incidence of difficulty accessing surgical care differed according to year (2005–2014).

In order to examine sociodemographic and surgery-related characteristics in the subsample of patients who required a past-year non-emergency surgery, we merged the individual (biannual/annual) CCHS data files to form a single aggregate dataset. Weighted frequencies determined the incidence of each type of difficulty accessing surgical care. Chi-square analyses and independent samples t-tests identified sociodemographic and surgery-related characteristic associated with difficulty accessing surgical care compared to no difficulty. A multivariable logistic regression examined independent associations between sociodemographic characteristics (i.e., age, sex, marital status, income, racial origin, urbanicity) and surgery-related characteristics (i.e., type of surgery, required overnight stay) and difficulty accessing surgical care, in a single model. All statistical analyses were conducted using SPSS and Stata statistical software [22,23] and we employed appropriate weighting and bootstrapping techniques (developed by Statistics Canada); the latter was included in the regression analysis for variance estimation.

## Results

### Sample characteristics

The annual incidence of past-year non-emergency surgery and any difficulty accessing surgical care from 2005 to 2014 are reported in Fig 1. Results of the chi-square analysis showed a significant difference in the incidence of difficulty accessing surgical care according to year ($X^2 = 83.50$, $p < .001$), ranging from 14.5% in 2005/2006 to 17.8% in 2014; the lowest incidence was in 2009 (12.6%) and the highest was in 2013 (22.8%). In the collapsed CCHS dataset for the combined years (weighted $N = 42,245,996$; includes only respondents who participated in the modules of interest), 7.2% (weighted $n = 3,052,072$) of Canadians reported that they required non-emergency surgery in the past-year. Among those who required non-emergency surgery, 15.6% experienced difficulty accessing surgical care.

### Types of difficulties accessing surgical care

Among those who required a past-year non-emergency surgery and experienced difficulty accessing surgical care, the most common types of difficulties endorsed were physician or systemic reasons, including: "waited too long for surgery" (58.5%); "difficulty getting an

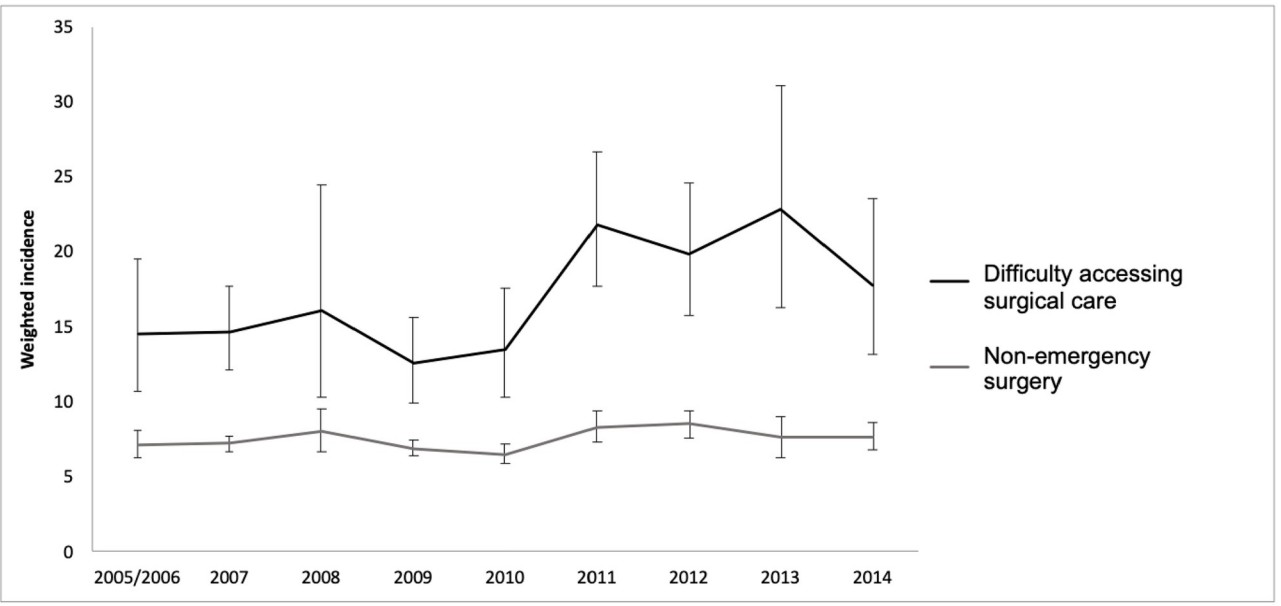

**Fig 1. Plotted weighted incidence of past-year non-emergency surgery and difficulties accessing surgical care.**

appointment" (30.7%); and "waited too long for a diagnostic test" (16.2%). Only 8.0% of respondents endorsed patient reasons for difficulty and 10.7% endorsed other reasons (Table 1).

## Sociodemographic disparities

The incidence of difficulty accessing surgical care significantly differed according to sex ($X^2$ = 4.02, $p$ = .045). In line with a higher prevalence of females undergoing non-emergency surgery compared to males (i.e., 55.0% versus 45.0%), those who endorsed eXperiencing difficulty were primarily female (52.4%). However, males had a higher incidence of difficulty (16.5%) compared to females (14.9%). No other significant differences in sociodemographic characteristics emerged (Table 2).

## Characteristics of surgery

The surgery-related characteristics are shown in Table 2. Differences in surgery type emerged between those who experienced difficulty and those who did not ($X^2$ = 96.09, $p$ < .001); "other" surgery (71.9%) and hip/knee surgery (11.4%) were the most common types of surgeries among those who experienced difficulty, whereas "other" surgery (69.4%) and cataract/eye surgery (12.2%) were the most common types among those who *did not* experience difficulty. Those who had hip/knee surgery had the highest incidence of difficulty (24.3%), whereas those who had cataract/eye surgery had the lowest incidence of difficulty (6.5%).

Those who experienced difficulty were also more likely to have required an overnight hospital stay compared to those who did not experience difficulty ($X^2$ = 15.39, $p$ < .001; 42.4% versus 28.2%); 16.3% of those who required an overnight stay endorsed experiencing difficulty. Those who experienced difficulty were also more likely to have their surgery cancelled or postponed by a surgeon ($X^2$ = 4.36, $p$ < .05; 52.4% versus 41.4%) or the hospital ($X^2$ = 9.86, $p$ < .05; 36.9% versus 21.7%), but less likely to cancel or postpone it themselves ($X^2$ = 19.01, $p$ < .001; 16.6% versus 42.6%). Across all surgery-related characteristics, those who had their surgery

**Table 1. Weighted incidence of reported reasons for difficulty accessing surgical care.**

|  | Past-year non-emergency surgery ($n$ = 3,052,072; 7.2%) |
| --- | --- |
|  | $n$(%) |
|  | Any difficulty accessing surgical care ($n$ = 476,371; 15.6%) |
| Physician/systemic reasons | 443,663 (93.2) |
| Waited too long for surgery | 278,824 (58.5) |
| Difficulty getting an appointment with a surgeon | 146,362 (30.7) |
| Waited too long for a diagnostic test | 77,250 (16.2) |
| Still waiting | 56,283 (11.8) |
| Difficulty getting a diagnosis | 49,907 (10.6) |
| Waited too long for a hospital bed | 46,423 (9.7) |
| Appointment cancelled/deferred | 36,405 (7.6) |
| Service not available | 20,190 (4.2) |
| Patient reasons | 38,190 (8.0) |
| Deterioration of health | 19,691 (4.1) |
| Cost | 14,608 (3.1) |
| Difficulty acquiring transportation | 5,474 (1.1) |
| Unable to leave house due to health | 1,836 (0.4) |
| Personal/family responsibility | 1,561 (0.3) |
| Language barrier | 400 (0.1) |
| Other reasons | 51,086 (10.7) |

Values represent weighted $n$(%); Participants were allowed to endorse more than one reason for experiencing difficulty accessing surgical care.

cancelled or postponed by the hospital experienced the highest incidence of difficulty accessing surgical care (42.1%).

## Correlates of difficulty accessing surgical care

Results of the multivariable logistic regression including sociodemographic and surgery-related characteristics are displayed in Table 3. Results revealed that requiring an overnight hospital stay was independently associated with increased odds of endorsing any difficulty accessing surgical care (adjusted odds ratio = 1.96, 95% confidence interval [1.17–3.29], $p <$ .05). No other characteristics were independently associated with difficulty accessing surgical care in the multivariable model.

## Waiting times

With respect to waiting times, those who experienced difficulty accessing surgical care were less likely to rate their waiting time as "acceptable" compared to those who did not experience difficulty ($X^2$ = 1,261.16, $p <$ .001; 37.4% versus 91.1%), despite indicating a higher mean acceptable waiting time in their particular case ($t$ = 4.40, $p <$ .001; $M$ = 53.93 days versus $M$ = 37.01 days, respectively); only 6.1% of those who rated their waiting time as "acceptable" endorsed experiencing difficulty. Those who experienced difficulty also had significantly longer waiting times compared to those who did not experience difficulty ($t$ = 10.59, $p <$ .001; $M$ = 130.83 days versus $M$ = 54.27 days).

**Table 2. Patient and surgery characteristics with and without difficulty accessing surgical care.**

| | Past-year non-emergency surgery (n = 3,052,072; 7.2%) | | | | | | Chi-square/t-test |
|---|---|---|---|---|---|---|---|
| | Experienced difficulty (n = 476,371; 15.6%) | | | Did not experience difficulty (n = 2,574,886; 84.4%) | | | |
| | *n* | % of difficulty[a] | % of characteristic[b] | *n* | % of no difficulty[a] | % of characteristic[b] | |
| Sociodemographic characteristics | | | | | | | |
| Age[c] | 50.79 | 0.02 | | 49.25 | 0.01 | | 0.38 |
| Sex | | | | | | | 4.02* |
| Male | 226,669 | 16.5 | 47.6 | 1,145,838 | 83.5 | 44.5 | |
| Female | 249,702 | 14.9 | 52.4 | 1,429,048 | 85.1 | 55.5 | |
| Household income | | | | | | | 0.16 |
| $29,999 or less | 76,300 | 16.8 | 22.8 | 377,927 | 83.2 | 22.6 | |
| $30,000–59,999 | 108,608 | 15.2 | 32.5 | 608,094 | 84.8 | 36.3 | |
| $60,000+ | 149,095 | 17.8 | 44.6 | 687,080 | 82.2 | 41.1 | |
| Racial origin | | | | | | | 0.08 |
| White | 379,739 | 15.1 | 84.8 | 2,142,637 | 84.9 | 87.1 | |
| Other | 67,981 | 17.6 | 15.2 | 318,374 | 82.4 | 12.9 | |
| Urbanicity | | | | | | | 0.96 |
| Urban | 375,164 | 15.3 | 78.8 | 2,075,692 | 84.7 | 80.6 | |
| Rural | 101,207 | 16.9 | 21.2 | 499,194 | 83.1 | 19.4 | |
| Marital status | | | | | | | 2.35 |
| Married/common-law | 315,382 | 15.8 | 66.2 | 1,677,898 | 84.2 | 65.3 | |
| Widowed/separated/divorced | 78,721 | 17.0 | 16.5 | 385,243 | 83.0 | 15.0 | |
| Single | 81,947 | 13.9 | 17.2 | 505,773 | 86.1 | 19.7 | |
| Surgery characteristics | | | | | | | |
| Type of surgery | | | | | | | 96.09*** |
| Hip/knee | 33,969 | 24.3 | 11.4 | 105,903 | 75.7 | 5.7 | |
| Cataract/eye | 15,952 | 6.5 | 5.3 | 227,834 | 93.5 | 12.2 | |
| Cancer | 15,538 | 12.6 | 5.2 | 108,063 | 87.4 | 5.8 | |
| Cardiac | 7,708 | 11.9 | 2.6 | 57,252 | 88.1 | 3.1 | |
| Gall bladder | 5,758 | 12.6 | 1.9 | 39,801 | 87.4 | 2.1 | |
| Hysterectomy | 5,094 | 12.7 | 1.7 | 34,944 | 87.3 | 1.9 | |
| Other | 214,857 | 14.2 | 71.9 | 1,300,515 | 85.8 | 69.4 | |
| Required overnight hospital stay | 99,571 | 16.3 | 42.4 | 509,600 | 83.7 | 28.2 | 15.39*** |
| Surgery cancelled or postponed by | | | | | | | |
| Surgeon | 39,945 | 35.1 | 52.4 | 73,747 | 64.9 | 41.4 | 4.36* |
| Hospital | 28,124 | 42.1 | 36.9 | 38,669 | 57.9 | 21.7 | 9.86** |
| Respondent | 12,621 | 14.2 | 16.6 | 76,202 | 85.8 | 42.8 | 19.01*** |
| Waiting time was acceptable | 109,399 | 6.1 | 37.4 | 1,694,317 | 93.9 | 91.1 | 1261.16*** |
| Acceptable wait time (days)[c] | 53.93 | 4.69 | | 37.01 | | | 4.40*** |
| Total time waited for surgery (days)[c] | 130.83 | 10.98 | | 54.27 | | | 10.59*** |

Values represent weighted *n*(%); Patients were allowed to endorse more than one reason for surgery cancellation.

[a]Represents the incidence of difficulty (or no difficulty) accessing surgery, according to the specified patient/surgery characteristic.

[b]Represents the prevalence estimates of each patient/surgery characteristic among those who did/did not experiencing difficulty accessing surgery.

[c]Values represent *M(SE)* and *t*-statistic.

*$p < .05$,

**$p < .01$,

***$p < .001$.

**Table 3. Associations between sociodemographic and surgery-related characteristics with any difficulty accessing surgical care.**

|  | Any difficulty |
| --- | --- |
|  | AOR (95% CI) |
| Age | 1.00 (0.99–1.01) |
| Sex |  |
| Male | 1.33 (0.87–2.03) |
| Female | 1.00 |
| Household income |  |
| $29,999 or less | 1.00 |
| $30,000–59,999 | 1.07 (0.66–1.71) |
| $60,000+ | 1.41 (0.83–2.39) |
| Racial origin |  |
| White | 0.48 (0.22–1.06) |
| Other | 1.00 |
| Urbanicity |  |
| Urban | 1.03 (0.72–1.48) |
| Rural | 1.00 |
| Marital status |  |
| Married/common-law | 1.00 |
| Widowed/separated/divorced | 0.90 (0.56–1.42) |
| Single | 0.71 (0.42–1.19) |
| Type of surgery |  |
| Hip/knee | 1.17 (0.63–2.17) |
| Cataract/eye | 0.58 (0.29–1.16) |
| Cancer | 0.46 (0.18–1.13) |
| Cardiac | 0.53 (0.14–1.95) |
| Gall bladder | 1.17 (0.46–2.98) |
| Hysterectomy | 0.49 (0.05–5.10) |
| Other | 1.00 |
| Required overnight hospital stay |  |
| Yes | 1.96 (1.17–3.29)* |
| No | 1.00 |

Reference categories are indicated by "1.00"; AOR = adjusted odds ratio (all characteristics entered into a single model); CI = confidence interval;

*$p < .05$.

## Discussion

The current study represents a descriptive examination of difficulty accessing non-emergency surgical care in Canada from 2005–2014. This study extends the work by the Fraser Institute through establishing the first Canadian population-based estimate of a wide variety of types of difficulties accessing surgical care over an extended duration and examining personal and surgery-related factors associated with difficulties accessing surgical care. Over 15% of individuals who required non-emergency surgery reported difficulty accessing that surgery, and rates of experiencing difficulty significantly increased across the study duration. Further, nearly 60% of individuals who experienced difficulty accessing surgical care cited a lengthy waiting time for their surgery. Those who endorsed difficulty waited on average 75 days longer for surgery than those who did not, in line with prior research indicating surgical patients commonly wait

longer than is recommended [24,25]. Finally, individuals who experienced difficulty were more likely to have their surgery cancelled or postponed by a surgeon or the hospital compared to those who did not experience difficulty.

The largest increase in difficulty accessing surgery occurred between 2010 (13.5%) and 2011 (21.8%). Since 2011, approximately 1 in 5 Canadians requiring non-emergency surgery experienced difficulty. Although speculative, this increase may relate to changes in health care expenditures, including a reduction in annual funding as part of the Wait Times Reduction Fund portion of Canada's 10-year Plan to Strengthen Health Care [17,26,27] and the Patient Wait Times Guarantee [28]. Both initiatives aimed to establish benchmarks for acceptable waiting times and allocate funds to increase healthcare resources, and both ended between 2010–2011. The termination of these initiatives may have influenced the increase in difficulty accessing surgical care beginning in 2011, considering the most common reason cited was "waited too long for surgery". The increase in difficulties experienced following the termination of these initiatives may highlight their potential efficacy. The incidence of difficulties also gradually increased across the study duration. Researchers suggest there is an increasing demand for healthcare associated with the growing aging population [29]. This may translate to greater burden on the healthcare system, and subsequently, increased endorsement of difficulties accessing care.

A difference emerged in the rates of difficulty accessing surgery according to sex. Despite the majority of non-emergency surgery patients being female, males had a higher incidence of difficulty accessing surgical care than females (i.e., 16.5% versus 14.9%). Research has shown that females have greater healthcare consumption [30], and males delay in seeking medical care [31]; as a result, males may experience greater difficulty accessing care. Differences may also be related to biases in self-report. A recent study showed that males experience less satisfaction after surgery compared to females, which may also contribute to perceptions of increased difficulty [32]. However, these hypotheses are speculative; future research should aim to understand factors that may be driving the emergent sex difference (e.g., healthcare utilization, type and complexity of surgery).

A greater proportion of Canadians who experienced difficulty accessing surgery required an overnight hospital stay compared to those who did not experience difficulty (42.4% versus 28.2%, respectively) and this was the only characteristic significantly associated with any difficulty accessing surgery in the multivariable model. A recent study of over 30,000 orthopedic surgery patients found the severity of preoperative health status predicted requiring an overnight stay [33]. This may suggest that individuals who experience difficulty have more severe health presentations preoperatively, and thus require additional healthcare resources (i.e., overnight hospital stay versus same-day discharge), which may impact access. Alternatively, experiencing difficulty accessing surgery may be associated with poorer postoperative health outcomes, and thus these patients might require additional medical attention after surgery, lengthening their hospital stay. In support, increased surgical wait time is associated with poorer postoperative outcomes including myocardial infarction, pneumonia, and mortality [34].

Results also demonstrated that "other" surgery and orthopedic (i.e., hip/knee) surgery were more prevalent among those who experienced difficulty and nearly one quarter of those who underwent orthopedic surgery endorsed difficulty accessing their surgery. This is in line with the Fraser Institute Canadian National Waiting List Survey, which found that across surgery types, there is the greatest disparity between the clinically reasonable wait time and actual wait time for orthopedic surgery (12.7 vs. 23.3 weeks, respectively), followed by neurosurgery (5.4 vs. 9.6 weeks) and head/neck surgery (8.3 vs. 11.4 weeks [35]); the latter two would be classified as "other" surgery in the current study.

Results should be considered alongside some limitations. First, although we utilized nationally representative data, only some provinces responded to the optional CCHS modules of interest; results may therefore not be generalizable and may be impacted by selection biases. Second, due to the cross-sectional nature of the data, causal inferences cannot be made. Third, this study utilized self-report data, which may be subject to response biases; however, research suggests that patient perceptions are robust and appropriate for health research [36]. Additionally, only past-year difficulties were examined and therefore are less susceptible to recall bias. Fourth, the sample excluded institutionalized individuals, Canadians living on reserves, and deceased individuals, which may have resulted in conservative estimates. Fifth, although we examined various types of surgeries and reasons for difficulty accessing surgical care, the range of categories included was limited according to the content of the CCHS. For example, we could not identify specific surgery types within each category, or explore additional types of difficulties (e.g., waiting for a medical device). Future research should examine a more comprehensive selection of surgery and difficulty types. Finally, the survey did not assess indicators of surgery risk or illness severity, which may have provided further insights as to why individuals endorse experiencing difficulty accessing surgical care.

Despite these limitations, the strengths of this study lie in its use of Canadian population-based data, spanning nearly a decade. This study outlines the increase in incidence of difficulty accessing surgical care over the course of approximately a decade and highlights the importance of targeted initiatives to improve patients' access to health services. Results provide insights toward the types of difficulties experienced by surgical patients and associated factors.

## Supporting information

**S1 Table. Provinces that completed the ACC and WTM modules according to survey cycle.** (DOC)

**S1 Appendix. Appendix A response rates.** (DOC)

**S2 Appendix. Appendix B access to health care services (ACC) and waiting times (WTM) questions.** (DOC)

## Acknowledgments

Although the research and analysis are based on data from Statistics Canada, the opinions expressed do not represent the views of Statistics Canada or the Canadian Research Data Centre Network (CRDCN).

## Author Contributions

**Conceptualization:** Eric Jacobsohn, Chris Christodoulou, Renée El-Gabalawy.

**Formal analysis:** Jordana Liyat Sommer, Edward Noh.

**Investigation:** Jordana Liyat Sommer, Edward Noh, Eric Jacobsohn, Chris Christodoulou, Renée El-Gabalawy.

**Methodology:** Jordana Liyat Sommer, Edward Noh, Renée El-Gabalawy.

**Resources:** Renée El-Gabalawy.

**Supervision:** Jordana Liyat Sommer, Eric Jacobsohn, Chris Christodoulou, Renée El-Gabalawy.

**Writing – original draft:** Jordana Liyat Sommer, Edward Noh.

**Writing – review & editing:** Jordana Liyat Sommer, Eric Jacobsohn, Chris Christodoulou, Renée El-Gabalawy.

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
