## [Decision Letter · Decision Letter 0]

13 Jul 2020

PONE-D-20-14535

An Examination of Difficulties Accessing Surgical Care in Canada from 2005-2014: Results from the Canadian Community Health Survey

PLOS ONE

Dear Dr. El-Gabalawy,

Thank you for submitting your manuscript to PLOS ONE. After careful consideration, we feel that it has merit but does not fully meet PLOS ONE’s publication criteria as it currently stands. Therefore, we invite you to submit a revised version of the manuscript that addresses the points raised during the review process.

Though only one reviewer out of many approached has agreed to review the paper, I agree with the comments provided. 

We look forward to receiving your revised manuscript.

Kind regards,

Itamar Ashkenazi

Academic Editor

PLOS ONE

Journal Requirements:

"This work was supported by University of Manitoba Start-Up Funding (El-Gabalawy), and the Max Rady College of Medicine and Rady Faculty of Health Sciences, University of Manitoba BSc(Med) program (Noh)."

"The authors received no specific funding for this work."

Reviewers' comments:

Reviewer's Responses to Questions

**Comments to the Author**

1. Is the manuscript technically sound, and do the data support the conclusions?

Reviewer #1: Partly

2. Has the statistical analysis been performed appropriately and rigorously? 

Reviewer #1: I Don't Know

3. Have the authors made all data underlying the findings in their manuscript fully available?

Reviewer #1: No

4. Is the manuscript presented in an intelligible fashion and written in standard English?

Reviewer #1: Yes

5. Review Comments to the Author

Reviewer #1: This well written manuscript examining the difficulties associated with accessing surgical care in Canada from 2005-2014. The study is strong as it presents a decade data from Canadian community health survey, however, is cross sectional and is limited by the nature of data (self -reported by patients). The authors analyzed data using Chi-square and t-test to show the differences in factors associated with difficulties accessing surgical care and ran a regression analysis of the factors in the final model.

They showed the factors (males, orthopedic surgery, and surgery cancelled by the surgeon or hospital) had the highest rates of difficulty, but not in their final regression model. However, they have provided a lengthy discussion on each individual factor.

For example, in page 11, authors stating that males have a higher incidence of difficulty accessing surgical care vs female, in spite that women are higher users of the health care system. The authors need to consider that the type of surgery may be different between male and female. Maybe it is easier to have hysterectomy compared to hip surgery (with the longest wait time). The regression model did not show any significant effect of Sex of difficulties accessing surgical care. This is true for other factors except staying overnight in hospital.

The discussion part is very lengthy, I suggest the authors make it more summarized and to the point.

6. PLOS authors have the option to publish the peer review history of their article (what does this mean?). If published, this will include your full peer review and any attached files.

Reviewer #1: No

---

## [Author Response · Author response to Decision Letter 0]

9 Sep 2020

Journal Requirements:

Response: We have now modified the manuscript formatting and file naming in accordance with the specified guidelines. 

"This work was supported by University of Manitoba Start-Up Funding (El-Gabalawy), and the Max Rady College of Medicine and Rady Faculty of Health Sciences, University of Manitoba BSc(Med) program (Noh)."

"The authors received no specific funding for this work."

Response: We have now removed funding information from the manuscript document (i.e., on the title page). We will change the Funding Statement within the online submission form to reflect the funding received from the Max Rady College of Medicine and Rady Faculty of Health Sciences, University of Manitoba BSc(Med) program for this project.

Response: There are restrictions on sharing a de-identified dataset, imposed by Statistics Canada. The CCHS data utilized in the current study may be accessed via submitting a proposal and receiving security clearance through a Canadian Research Data Centre (this is now clarified in the methods section). Restrictions are in place to protect the privacy of survey respondents. Please see application and guidelines website for information on applying to access Research Data Centre protected data.

Additional Clarifications:

Thank you for requesting to update your Financial Disclosure:

"This work was supported by Max Rady College of Medicine and Rady Faculty of Health Sciences, University of Manitoba BSc(Med) program."

We note the following funder that was mentioned in your manuscript is not included in this statement:

"This work was supported by University of Manitoba Start-Up Funding (El-Gabalawy)."

Please explain why this funder was removed from your funding disclosure statement and whether you did indeed receive funding from them during the course of your study. If they were a source of funding for your study, please confirm this and we will add them to your Financial Disclosure statement.

Response: We initially removed the University of Manitoba Start-Up Funding (El-Gabalawy) as it was unclear whether or not to include funding received by an investigator that is not specific to this study. The BSc(Med) funding was allocated specifically to support the current study, which is why that funding source was retained.

Please add the University of Manitoba Start-Up Funding back in. 

The funding statement should now read, 

“This work was supported by University of Manitoba Start-Up Funding (El-Gabalawy) and Max Rady College of Medicine and Rady Faculty of Health Sciences, University of Manitoba BSc(Med) program.”

Reviewer 1

Reviewer #1: This well written manuscript examining the difficulties associated with accessing surgical care in Canada from 2005-2014. The study is strong as it presents a decade data from Canadian community health survey, however, is cross sectional and is limited by the nature of data (self -reported by patients). The authors analyzed data using Chi-square and t-test to show the differences in factors associated with difficulties accessing surgical care and ran a regression analysis of the factors in the final model. They showed the factors (males, orthopedic surgery, and surgery cancelled by the surgeon or hospital) had the highest rates of difficulty, but not in their final regression model. However, they have provided a lengthy discussion on each individual factor. 

For example, in page 11, authors stating that males have a higher incidence of difficulty accessing surgical care vs female, in spite that women are higher users of the health care system. The authors need to consider that the type of surgery may be different between male and female. Maybe it is easier to have hysterectomy compared to hip surgery (with the longest wait time). The regression model did not show any significant effect of Sex of difficulties accessing surgical care. This is true for other factors except staying overnight in hospital.

Response: Thank you for your positive feedback. Regarding the sex difference, although it is possible this could be influenced by type of surgery, we included type of surgery in our multivariable regression and it did not emerge as a significant correlate of endorsing difficulty accessing surgical care. However, we have now added to our discussion on this matter. Page 19 now reads,

“future research should aim to understand factors that may be driving the emergent sex difference (e.g., healthcare utilization, type and complexity of surgery)”

The discussion part is very lengthy, I suggest the authors make it more summarized and to the point.

Response: We have now shortened the length of the discussion. 

Sincerely,

Renée El-Gabalawy PhD 

Assistant Professor at the University of Manitoba

Departments of Anesthesiology, Perioperative and Pain Medicine and Clinical Health Psychology

E-mail: renee.el-gabalawy@umanitoba.ca

---

## [Editor Report · Decision Letter 1]

21 Sep 2020

An Examination of Difficulties Accessing Surgical Care in Canada from 2005-2014: Results from the Canadian Community Health Survey

PONE-D-20-14535R1

Dear Dr. El-Gabalawy,

We’re pleased to inform you that your manuscript has been judged scientifically suitable for publication and will be formally accepted for publication once it meets all outstanding technical requirements.

Kind regards,

Itamar Ashkenazi

Academic Editor

PLOS ONE
---

## [Editor Report · Acceptance letter]

25 Sep 2020

PONE-D-20-14535R1 

An examination of difficulties accessing surgical care in Canada from 2005-2014: Results from the Canadian Community Health Survey 

Dear Dr. El-Gabalawy:

I'm pleased to inform you that your manuscript has been deemed suitable for publication in PLOS ONE. Congratulations! Your manuscript is now with our production department. 

Kind regards, 

on behalf of

Dr. Itamar Ashkenazi 

Academic Editor

PLOS ONE